# The impact of metastatic sites in advanced pancreatic adenocarcinoma, systematic review and meta-analysis of prospective randomized studies

**Pedro Luiz Serrano Usón, Junior**[1,2]*, **Fernanda D'Avila Sampaio Tolentino**[1], **Vanessa Montes Santos**[1], **Edna Terezinha Rother**[1], **Fernando Cotait Maluf**[1,3]

**1** Hospital Israelita Albert Einstein, Oncology Department, São Paulo, Brazil, **2** Mayo Clinic, Phoenix, Arizona, United States, **3** Centro Oncológico Antônio Ermírio de Moraes, Beneficência Portuguesa de São Paulo, São Paulo, Brazil

* pedroluiz_uson@hotmail.com

**Data Availability Statement:** All relevant data are within the manuscript and its Supporting Information files.

## Abstract

The real impact of specific sites of metastasis on prognosis of metastatic pancreatic cancer (MPC) is unknown. To evaluate the association of specific metastatic sites and survival outcomes in MPC a systematic literature review was performed including prospective randomized trials of systemic treatments in metastatic pancreatic cancer indexed in PubMed, Embase and Web of Science. Data regarding systemic treatment regimens, progression free survival and overall survival were extracted. The outcomes were compared using a random effects model. The index I2 and the graphs of funnel plot were used for the interpretation of the data. Of 1,052 abstracts, 7 randomized trials were considered eligible with a combined sample size of 2,975 MPC patients. Combining the studies with meta-analysis, we could see that patients with liver metastasis had a HR for death of 1.53 with 95% CI of 1.15 to 2.02 (p-value 0.003) and HR for risk of progression of 1.96 with 95% CI of 1.28 to 2.99 (p-value 0.002), without significant heterogeneity. Having two or more sites of metastasis comparing to one site did not have impact on overall survival; RR of 1.05 with 95% CI 0.91 to 1.23 (p-value 0.493). In conclusion, liver metastasis confers worse outcomes among patients with MPC. Apparently, multiple metastatic sites do not present worse prognosis when compared with only one organ involved, therefore, demonstrating the severity of this disease. Prospective studies evaluating other treatments are necessary to address the impact of local treatments in liver metastasis in MPC.

## Introduction

Pancreatic cancer (PC) is a deadly cancer, being among the five most lethal malignant tumors in the last years [1]. Most patients are diagnosed with advanced disease and few survival gains were obtained in the past decades [2].

Efforts have been made to obtain better outcomes. Lately, numerous studies have described molecular advances to improve and develop guided therapies [3–5]. However, the applicability of molecular classifications in treatment decisions is still a topic to be evaluated [6].

**Funding:** The authors received no specific funding for this work.

**Competing interests:** The authors have declared that no competing interests exist.

A deep understanding about the disease and its particularities is fundamental, in terms of outcomes and prognosis. Based on the current literature, the prognostic relationship with some factors—such as pre and postoperative CA19-9 serum levels, lymph node involvement and tumor size—are already well established [7,8].

Criteria such as volume of disease and number of metastatic sites are related to outcome in other tumor subtypes, such as prostate cancer [9]. In metastatic pancreatic cancer (MPC), there also appears to be an association between outcomes and specific metastasis. There are research groups reviewing this question and suggesting stratification of metastasis by different sites, however, few studies have approached this subject in the published literature [10].

To elucidate these questions, this study aims to perform a systematic review and meta-analysis of randomized trials that collected data from metastasis sites and evaluate the association of specific metastatic sites and survival outcomes in MPC.

## Materials and methods

### Search strategy

This study was designed in conformity with the 2009 Preferred Reporting Items for Systematic Reviews and Meta-Analysis (PRISMA) statement guidelines [11]. We searched PubMed (1950–2019) on July 2019. We used the keywords ((((((((((((pancreatic neoplasms [MeSH Terms]) OR (pancreatic neoplasms[Title/Abstract] OR Pancreatic cancer[Title/Abstract] OR pancreatic tumor*[Title/Abstract] OR pancreatic tumour* [Title/Abstract] OR pancreas neoplasms[Title/Abstract] OR pancreas cancer[Title/Abstract] OR pancreas tumor*[Title/Abstract] OR pancreas tumour* [Title/Abstract])) OR (pancreatic neoplasms[Text Word] OR Pancreatic cancer[Text Word] OR pancreatic tumor* [Text Word] OR pancreatic tumour* [Text Word] OR pancreas neoplasms[Text Word] OR pancreas cancer[Text Word] OR pancreas tumor* [Text Word] OR pancreas tumour* [Text Word])) OR pancreatic ductal adenocarcinoma [Title/Abstract]) OR pancreatic ductal adenocarcinoma [Text Word]) OR "cancer of the pancreas" [Text Word]) and Best Match Filters: Clinical Trial, Phase III; with 196 abstracts. We searched Embase and Web of Science using the following keywords (((('pancreas'/mj AND cancer* OR pancreatic) AND cancer* OR pancreatic) AND neoplasm* OR pancreatic) AND adenocarcinoma* OR pancreatic) AND tumor* AND [randomized controlled trial]/lim. Results retrieved more 926 abstracts. After excluding duplications and adding 5 more records identified through other sources (google search), the final database sample was made up of 1,052 records (**Fig 1**). The protocol (number 3493–18) was registered in the system for research project management (SGPP) of Hospital Israelita Albert Einstein. The protocol is available for consultation upon request.

Two authors (P.L.S.U.J. and V.M.S.) reviewed all abstracts. Inclusion criteria were: (1) randomized prospective cohorts investigating first-line systemic treatment; (2) patients with metastatic pancreatic adenocarcinoma; (3) available information of metastatic sites. Exclusion criteria were: (1) language other than English; (2) duplicate publication; (3) review articles, and case reports; (4) clinical trial protocols.

After a preliminary review, 133 full-text articles were selected for evaluation (**Fig 1**). Reasons for exclusion were: non-randomized study (n = 43), review articles and meta-analysis (n = 215), experimental research (n = 118), non-first-line therapy (n = 34), non-pancreatic adenocarcinoma (n = 224), resectable disease/ locally advanced (n = 237) and clinical trial protocol (n = 48).

After review of full-text articles, we excluded 126 articles. Reasons for exclusions were no data about metastatic sites (n = 88) and duplicates (n = 38). **Seven articles were selected for meta-analysis.**

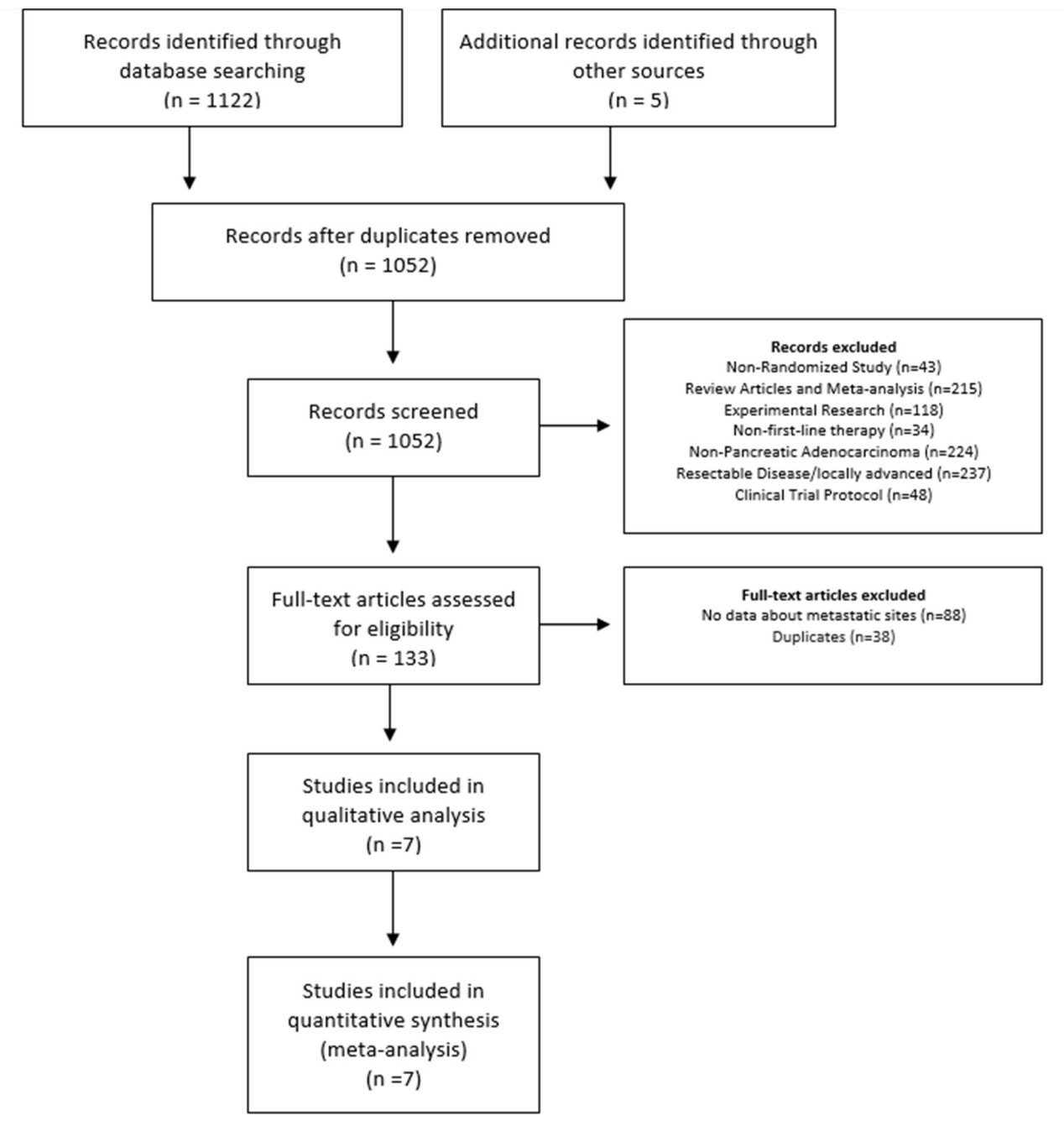

**Fig 1. Flow Diagram of the included studies.**

## Definitions and outcomes of interest

Two authors (P.L.S.U.J. and F.T.) extracted data of all included studies using a standardized data collection form. Primary outcome was overall survival (OS) related to metastatic sites. Secondary outcome was evaluation of progression free survival (PFS). Data were collected as published in the studies. Nonrandomized retrospective and phase II studies were not included in this analysis because they have a higher risk of bias than randomized studies and the resultant larger sample size with just randomized data provides greater reliability.

## Data extraction, data synthesis and analysis

For meta-analysis we considered articles that met the inclusion criteria and presented a hazard ratio (HR) followed by standard errors / confidence intervals or medians of survival of the groups accompanied by the number of events in each group (to allow the estimation of variability). For studies that presented HR and CI, we estimated the log (HR) by log HR transformation and the standard error (SE) was estimated as (log (upper limit of the CI)—log (Lower limit of the CI)) / 3.92. For the studies that we don't have CI, the confidence interval limits will then be given by HR—1.96 x SE (HR) and HR + 1.96 x SE (HR), the comparison of patients with liver metastasis was performed with patients with metastasis elsewhere.

For studies that presented the median survival of two or more groups, and the number of events in each group, we estimated the risk ratio (RR) for each group compared with a group defined as a reference [12].

We combined data using a random-effects-model meta-analysis, because we assumed that results of different studies depended not only on the sample variation and the covariables investigated, but also on other factors. The I2 index was used to measure the heterogeneity of the results of the different studies. Publication bias was evaluated by funnel plot and analyses were performed using the R [13] and Meta [14] packages, considering a significance level of 5%.

## Quality assessment

Methodological quality assessments of the included studies were accessed using the Revised Cochrane risk-of-bias tool for randomized trials. The scale is constituted by five domains, namely (1) bias arising from the randomization process; (2) bias due to deviations from intended interventions; (3) bias due to missing outcome data; (4) bias in measurement of the outcome; (5) bias in selection of the reported result. Each domain is judged as low risk, some concerns or high risk of bias, two authors (P.L.S.U.J. and V.M.S.) have done the quality assessment [15].

## Results

All seven studies were published after 2009. Sample size ranged from 125 to 607 patients, with a combined sample of 2,975 patients. Almost all included studies evaluated gemcitabine-based chemotherapy regimens [16–21]. Only the study from Conroy et al. evaluated FOLFIRINOX [22]. Two randomized phase 2 [16,19] and five randomized phase 3 studies [17,18,20–22] presented data related with the presence or absence of liver metastasis only, in metastatic patients, two studies also reported data (number of events) on lung metastasis and comparative outcomes of 1 or more metastatic sites [21, 22], one study reported results of lymph node metastasis [17], and one about peritoneal metastasis [21] (**Table 1**). The studies were evaluated using the Revised Cochrane risk-of-bias tool for randomized trials (**Fig 2**), details about the assessment in each trial and other information's can be found in **S1 and S2 Tables**.

## Liver metastasis

The HR for death in the presence of liver metastasis ranged in the studies between 0.83 (0.68; 1.02) -2.36 (1.63; 3.42), only one study did not have the hazard for death [22]. Comparing the six studies (those that presented outcome data related to each medication arm were included as individual data in meta-analysis, i.e. Fuchs et al. 2015 and Fuchs et al. 2015.1. refers to study arms 1 and 2) including 2,633 patients, we had a global estimate for HR of 1.53 with 95% CI of 1.15 to 2.02 (p-value 0.003), without significant heterogeneity (I2 = 5.03%) (**Fig 3**). Regarding

**Table 1. Summary of included studies.**

| Study | Type | Treatment | N° | Data on metastasis | | |
|---|---|---|---|---|---|---|
| | | | | Liver | Lung | Sites |
| Borad et al. (2015) [16] | Phase II | Gemcitabine vs Gemcitabine plus TH-302 | 214 | Y | N | N |
| Conroy et al. (2011) [22] | Phase III | Gemcitabine | 171 | Y | Y | Y |
| | | FOLFIRINOX | 171 | Y | Y | Y |
| Deplanque et al. (2015) [17] | Phase III | Gemcitabine vs Gemcitabine plus masitinib | 348 | Y | N | N |
| Fuchs et al. (2015) [18] | Phase III | Ganitumab 12 mg/kg plus Gemcitabine | 318 | Y | N | N |
| | | Ganitumab 20 mg/kg plus Gemcitabine | 160 | Y | N | N |
| Kindler et al. (2012) [19] | Phase II | Gemcitabine plus Ganitumab 12 mg/kg or Conatumumab 10mg/kg or placebo | 125 | Y | N | N |
| Van Cutsem et al. (2009) [20] | Phase III | Gemcitabine plus erlotinib vs Gemcitabine plus erlotinib plus bevacizumab | 607 | Y | N | N |
| Von Hoff et al. (2013) [21] | Phase III | Gemcitabine | 430 | Y | Y | Y |
| | | Gemcitabine plus Nab-paclitaxel | 431 | Y | Y | Y |

N°: Number of patients; TH-302: Evofosfamide; FOLFIRINOX: fluorouracil, irinotecan and oxaliplatin; Y: Yes; N: No

publication bias, the funnel plot (**Fig 4**) shows a lack of studies with more patients than the polled average. Unfortunately, only seven of randomized trials had objective data on metastasis outcomes. However, in a sample of more than 2,500 patients, the inclusion of more studies would have a little effect in the outcome of this analysis.

Three studies reported the presence of liver metastasis and PFS [16,18,19]. With a combined sample size of 817 patients, a global estimation by meta-analysis obtained a HR for relapse risk (progression) of 1,96 with 95% CI of 1.28 to 2.99 (p-value 0.002). The I2 index was 0.00% (**S1 Fig**).

## Lymph nodes, lung and peritoneum

The analysis of other metastatic sites such as lymph nodes, lungs and peritoneum were not performed because just one study included HR for OS for lymph nodes [17], and one study reported data of mean OS for lung metastasis [21]. It is important to mention that the HR for OS with lung metastasis in the Conroy et al. [22] study was calculated by authors (by number of events) with a HR of 0.89; CI 95% of 0.78 to 1.02 (p-value 0,089), but no statistical significance was found.

## Number of metastatic sites

For the presence of two, three or more metastasis sites in relation to one regarding death risk, two studies reported numbers of events [21, 22]. Interestingly, these studies considered regimens that are currently used in first line setting (FOLFIRINOX and Gemcitabine-Nab-Paclitaxel). None of the combined analyzes of the two studies compared two, three or more metastasis sites with one site only presented by statistically significant results, i.e., RR of 1.10 with 95% CI of 1.00 to 1.21 (0.061) I2 index 0% and RR of 1.05 with 95% CI 0.91 to 1.23 (p-value 0.493) I2 index 43.10% (**Fig 5**). These results suggested that number of different metastasis sites have no impact on the overall survival outcome for patient with metastatic pancreatic cancer.

## Discussion

This meta-analysis of prospective randomized studies showed that patients with liver metastasis in MPC have worse outcomes (PFS and OS). The analysis of number of metastatic sites did not show any association.

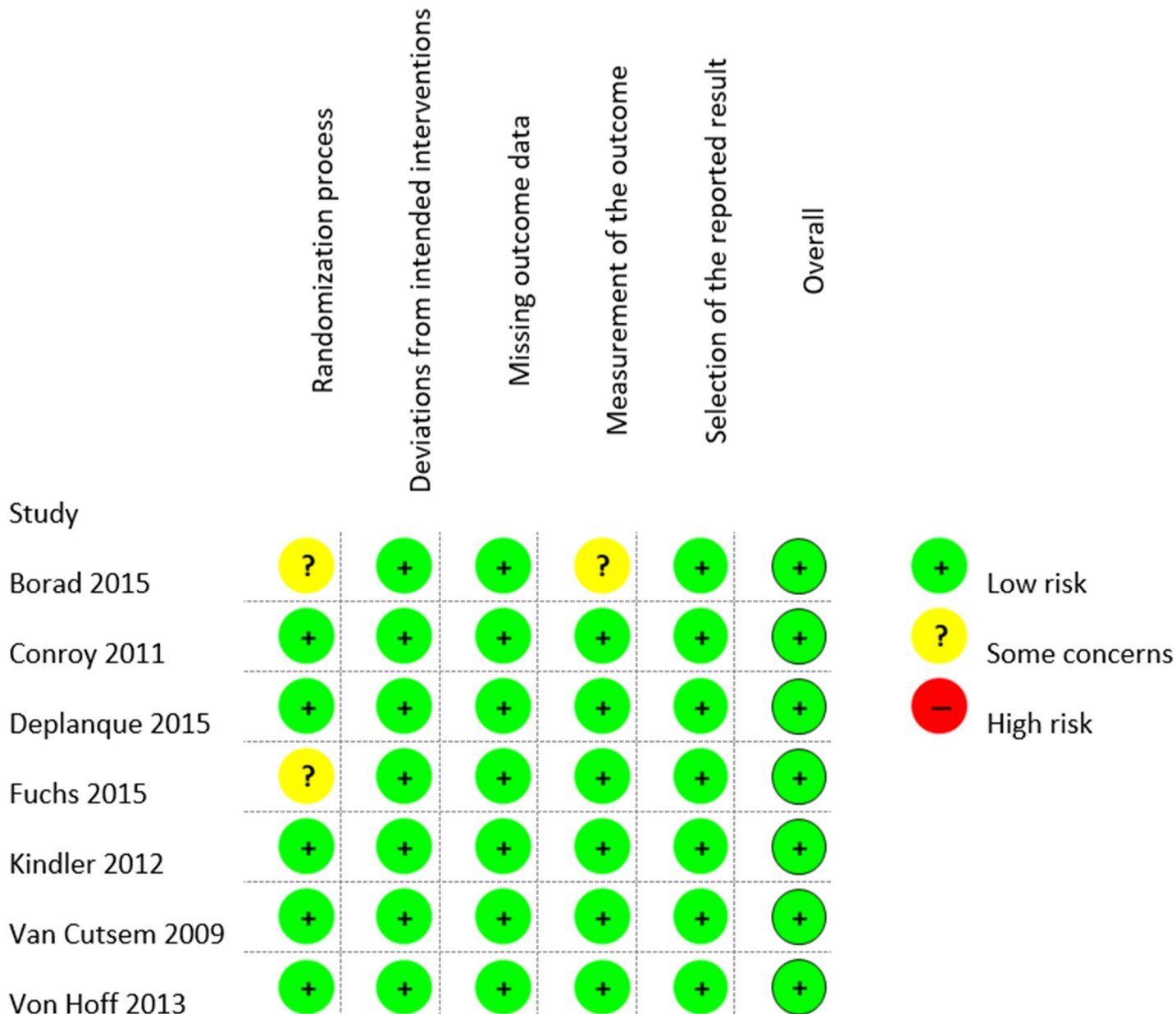

**Fig 2. Summary of risk of bias in the randomized controlled trials included.**

Several tumors have different behavior and outcomes depending on the involved metastatic sites. In breast cancer cases, visceral involvement is associated with worse outcomes compared with bone involvement [23]. The same is seen in prostate cancer in which randomized trials are designed to define low- or high-volume of metastatic disease and outcomes [24,25]. Most of the international series evaluating MPC have demonstrated that OS for most patients is less than one year, and classifications with respect to metastatic sites are not currently performed, probably because there are poor outcomes [1]. However, trends in reduced mortality are being seen [26].

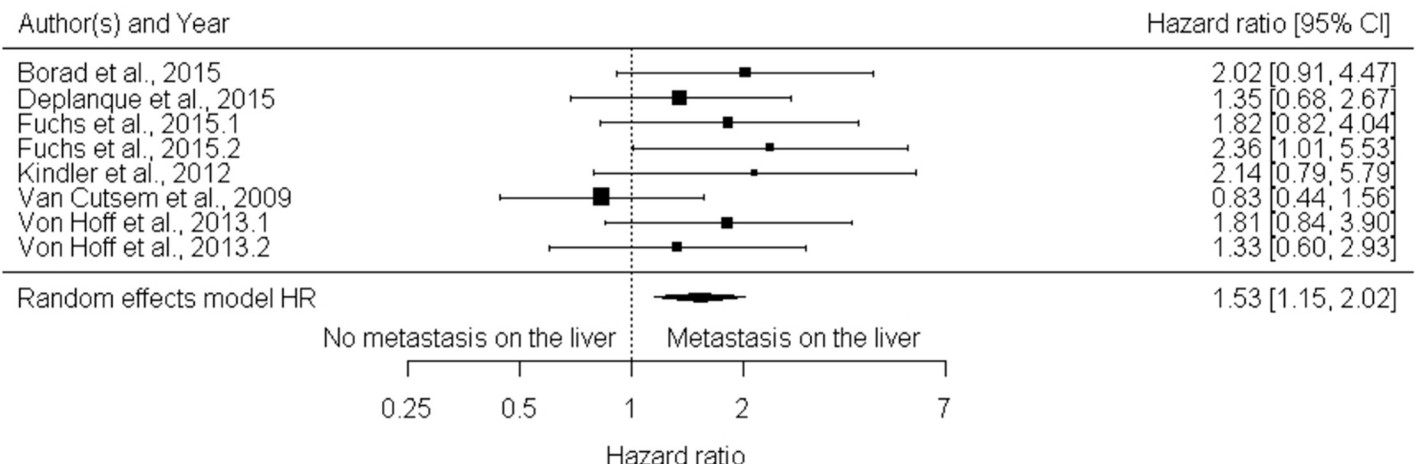

**Fig 3. Forest Plot for risk of death and liver metastasis based on hazard ratios (N = 2633).** Overall the presence of hepatic metastasis increases the risk of death around 50% (HR1,53 CI 95% [1,15; 2,02] (p 0,003).

An extensive systematic literature review was performed by Cannistra and colleagues to determine the impact and outcomes of metastasis in pancreatic cancer [10]. Most studies

**Fig 4. Funnel plot for HR of OS and liver metastasis.** Funnel plot shows the absence of studies with a greater number of patients than the mean analyzed.

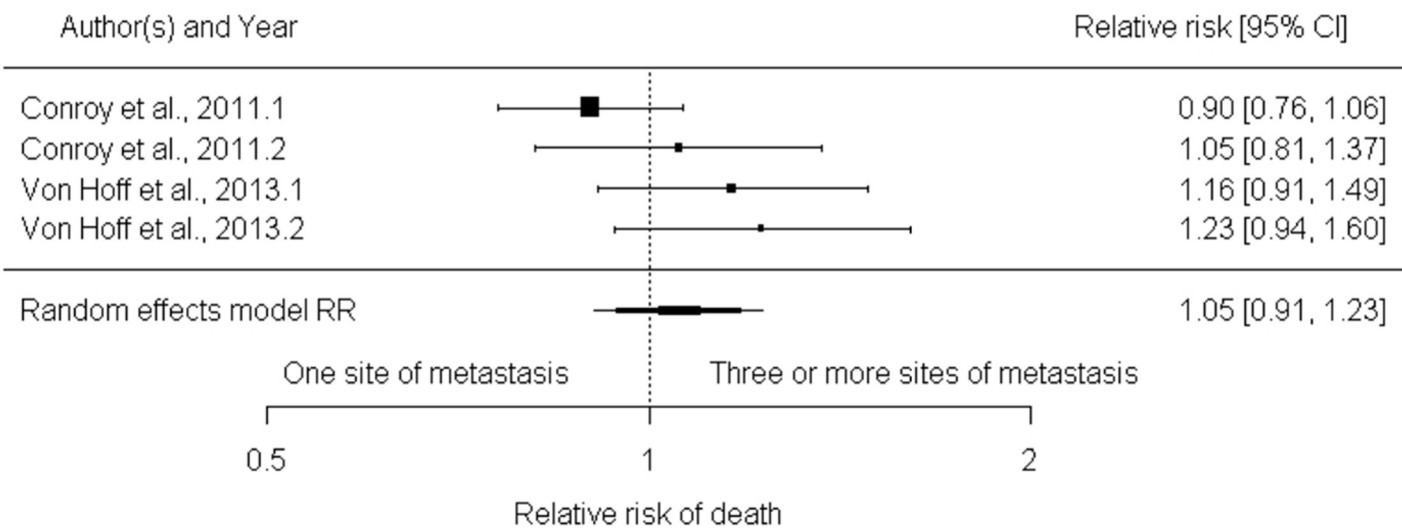

**Fig 5. Forest plot for relative risk of death with three or more sites of metastasis compared to one site.** The forest plot based on number of events, shows absence of statistical significance of the sites of metastasis in relative risk of death.

included in the analysis were case series and reports, and few randomized data were included. They concluded that liver metastasis provides better outcomes than metastasis from other sites, and particularly better outcomes with isolated liver metastasis. Of note is that, most of the included studies evaluated local treatments as resection and given that journals often favor positive-outcomes findings the risk of publication bias should be also considered.

Liver is one of the major metastatic sites in gastrointestinal cancers. Randomized trials have addressed resection combined with perioperative chemotherapy in colorectal cancer and have reported better outcomes using these strategies. Several case series have pointed out benefit of locoregional treatments in liver metastasis in MPC, but randomized data on the subject are scarce [10,27]. Our results show that liver metastasis confer a HR for death of 1.53 with 95% CI of 1.15 to 2.02 (p-value 0.003), without significant heterogeneity (I2 = 5.03%) and a HR for risk of relapse of 1.96 with 95% CI of 1.28 to 2.99 (p-value 0.002). Based on these poor outcomes among patients with metastatic disease and the studies previously discussed, locoregional strategies including resection may be an option for future trials.

In this systematic review, 133 randomized trials were included, but only seven had individual metastatic sites related with data outcomes. Despite the exclusion of most of studies, the analysis could be performed with a sample of more than 2,900 randomized patients, strengthening the results that were found. Other relevant fact is the inclusion in the analysis of two positive chemotherapy randomized phase 3 trials with regimens that are used in clinical practice [21,22]. These findings demonstrate a worse outcome of liver metastasis compared with other sites in patients treated with modern chemotherapy regimens.

A consensus statement from a group of experts from the pancreatic cancer field provided mandatory and recommended baseline measurements and prognostic characteristics to be included in trials investigating palliative systemic therapy for MPC. Sites and number of metastatic sites in MPC were considered mandatory variables, and these mandatory prognostic factors should be included in regression analyses to adjust their effect on treatment outcomes [28]. This recommendation provides strength to our findings and reinforces the need for a better characterization of the impact of sites and number of metastatic sites in MPC in outcomes of randomized trials. The adequacy of future clinical trials evaluating MPC could include

analyses of non-conventional metastatic sites such lungs, bones and central nervous system. The analyses of the impact of these sites were not possible to perform in our study because few studies considered regression data from these sites.

Publication bias in this study was evaluated by the funnel plot. We can observe that largest studies with small standard errors, placed at the top of the graph were absent. This bias could be overcome by the inclusion of more randomized trials, but as previously discussed most of studies did not present regression models regarding metastatic sites. It is worth to emphasize that funnel plot is more appropriate for reviews with more than ten studies, but the inclusion of more patients would hardly change the results found.

This meta-analysis has some limitations, publication bias and lack of patient-level data. Considering that we included only published data, we clearly observed in the analysis of funnel plot the lack of studies, which lead to the pattern found. Publication bias is intrinsic to this type of study, but such bias becomes less relevant because of the number of patients included in the analysis and the quality of the studies included. Some trials also included locally advanced disease, but we used for hazard ratio comparison liver metastasis versus overall metastasis. The randomization of the original studies included in this analysis was performed regarding the treatment. We are not considering two treatments comparisons in this meta-analysis, but the real impact of different systemic treatments would hardly change the results found since no systemic treatment confers more than 10–12 months of overall survival in any randomized trial for metastatic pancreatic adenocarcinoma. Finally, individual data of patients could have helped to identify other sources of heterogeneity and the inclusion of more studies in the analysis, particularly in analysis of other metastatic sites individually, volume of liver disease and performance status (most of the randomized trials included just patients with good performance status). Analysis of other metastatic sites such as lymph nodes, lungs and peritoneum was not possible because all the studies evaluated did not include individual outcomes of these sites to draw exact conclusions of their real impact on patient outcomes, in addition, lymph node and peritoneal metastasis usually is accompanied by other sites of metastasis and multiple site analysis was possible, with no statistically significant difference found. An analysis of large survival series of pancreatic cancer patients may help to stratify the impact of performance status and other sites of metastasis in MPC. The strength of our analysis is related to the inclusion of a fair number of prospective well-designed studies with a combined sample size of more than 2,900 patients, and report of consistent findings.

The absence of association of more than two metastatic sites compared with one site outcomes shows that MPC is an aggressive disease that requires incorporation of more strategies for better stratification.

## Conclusion

Liver metastasis confers worse outcomes (PFS and OS) in patients with MPC. Apparently multiple sites of metastasis do not provide worse prognosis when compared with only one organ. Prospective randomized studies evaluating other treatments such as liver-directed therapies are warranted.

## Supporting information

**S1 Fig. Forest plot for relative risk of progression due to liver metastasis.**
(TIF)

**S1 Table. Summary of included studies.**
(XLSX)

**S2 Table. Summary of quality assessment.**
(XLSX)

**S3 Table. PRISMA checklist.**
(DOC)

## Acknowledgments

We gratefully acknowledge the statistical support given by the Research Support Center (NAP) of the Hospital Israelita Albert Einstein.

## Author Contributions

**Conceptualization:** Pedro Luiz Serrano Usón, Junior, Vanessa Montes Santos, Fernando Cotait Maluf.

**Data curation:** Fernanda D'Avila Sampaio Tolentino, Vanessa Montes Santos, Edna Terezinha Rother.

**Formal analysis:** Pedro Luiz Serrano Usón, Junior, Fernanda D'Avila Sampaio Tolentino, Edna Terezinha Rother.

**Methodology:** Pedro Luiz Serrano Usón, Junior, Vanessa Montes Santos, Edna Terezinha Rother.

**Project administration:** Fernando Cotait Maluf.

**Software:** Edna Terezinha Rother.

**Supervision:** Edna Terezinha Rother.

**Writing – original draft:** Pedro Luiz Serrano Usón, Junior, Fernanda D'Avila Sampaio Tolentino.

**Writing – review & editing:** Vanessa Montes Santos, Fernando Cotait Maluf.

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
