## [Decision Letter · Decision Letter 0]

6 Jan 2020

PONE-D-19-32606

The impact of metastatic sites in advanced pancreatic adenocarcinoma, systematic review and meta-analysis of prospective randomized studies

PLOS ONE

Dear Dr. Pedro Uson Junior,

Thank you for submitting your manuscript to PLOS ONE. After careful consideration, we feel that it has merit but does not fully meet PLOS ONE’s publication criteria as it currently stands. Therefore, we invite you to submit a revised version of the manuscript that addresses the points raised during the review process.

ACADEMIC EDITOR: I have comments for this article as follows:

1. Please consider that the outcome from first-line and second-line studies might be different.

2. In figure 2, all included studies show no impact of survival, but the final HR was a positive impact of liver metastasis to the overall survival? How come? Please explain. Also, why the authors have Fuchs et al. 2015 and Fuchs et al. 2015.1. There is no explanation in the text and analysis. Do they refer to study arms 1 and 2? Please clearly describe them in the manuscript. 

3. The legend of Figure 2 illustrates an HR 1.53 (1.15-2.02, p=0.003), which seems different from the figure itself (0.42[0.14-0.70]). Please explain.  

4. There were no clear abbreviations in the legend for some short words in the figures. Please correct them. 

5. Liver metastasis includes huge liver (higher %) metastasis and tiny liver metastasis (lower % of occupying lesions). The study failed to exclude the factor, resulting in a not scientific enough conclusion. Please at least declare the limitations of the study.   

Please respond to these questions as well. 

We would appreciate receiving your revised manuscript by Feb 20 2020 11:59PM. To enhance the reproducibility of your results, we recommend that if applicable you deposit your laboratory protocols in protocols.io, where a protocol can be assigned its own identifier (DOI) such that it can be cited independently in the future. For instructions see: http://journals.plos.org/plosone/s/submission-guidelines#loc-laboratory-protocols

We look forward to receiving your revised manuscript.

Kind regards,

Jason Chia-Hsun Hsieh, M.D. Ph.D

Academic Editor

PLOS ONE

2. We noticed you have some minor occurrence(s) of overlapping text with the following previous publication(s), which needs to be addressed:

https://doi.org/10.1038/s41598-019-52334-y

https://doi.org/10.1016/j.clcc.2018.03.007

In your revision ensure you cite all your sources (including your own works), and quote or rephrase any duplicated text outside the Methods section. Further consideration is dependent on these concerns being addressed."

Additional Editor Comments (if provided):

I have comments for this article as follows:

1. Please consider that the outcome from first-line and second-line studies might be different.

2. In figure 2, all included studies show no impact of survival, but the final HR was a positive impact of liver metastasis to the overall survival? How come? Please explain. Also, why the authors have Fuchs et al. 2015 and Fuchs et al. 2015.1. There is no explanation in the text and analysis. Do they refer to study arms 1 and 2? Please clearly describe them in the manuscript.

3. The legend of Figure 2 illustrates an HR 1.53 (1.15-2.02, p=0.003), which seems different from the figure itself (0.42[0.14-0.70]). Please explain.

4. There were no clear abbreviations in the legend for some short words in the figures. Please correct them.

5. Liver metastasis includes huge liver (higher %) metastasis and tiny liver metastasis (lower % of occupying lesions). The study failed to exclude the factor, resulting in a not scientific enough conclusion. Please at least declare the limitations of the study.

Please respond to them as well.

Reviewers' comments:

Reviewer's Responses to Questions

**Comments to the Author**

1. Is the manuscript technically sound, and do the data support the conclusions?

Reviewer #1: Yes

Reviewer #2: No

2. Has the statistical analysis been performed appropriately and rigorously? 

Reviewer #1: Yes

Reviewer #2: I Don't Know

3. Have the authors made all data underlying the findings in their manuscript fully available?

Reviewer #1: Yes

Reviewer #2: No

4. Is the manuscript presented in an intelligible fashion and written in standard English?

Reviewer #1: Yes

Reviewer #2: Yes

5. Review Comments to the Author

Reviewer #1: Comment 1: Statistically, this is a well executed meta-analysis. Moreover, the data from this manuscript may help in designing future trials and adjusting randomization by metastatic site in accrual (knowing survival biases in patients assignment), indeed PC with lung metastasis & bone metastasis are at lower death risk than those with liver metastasis.

Comment 2: Is the HR of "liver metastasis" versus the "overall metastasis" (any kind considered) and/or "locally advanced" as comparator? This is not well stated in the manuscript. Could you please clearly state the comparator to the "liver metastasis". Also, considering that "locally advanced" patients are included, it probably would be fair to mention this in the limitations.

Comment 3: line 153; What do you mean by "PE" (is it Standard Error)? Also, on line 153 you mention that you estimate the "Confidence Interval limits", but you mention on line 148-149 that confidence intervals are supplied by the original articles. Why do say, that you you re-estimate the Confidence Intervals, could you please explain?

Comment 4: Why do you use different scales to report the "HRs & 95%CI", between the "main text" and the Forest Plots (Figure 2, Figure 4 & Supplementary 3). For the Forest Plots (Figure 2, Figure 4 & Supplementary 3) you use a "logarithmic scale" which is not stated for for Figure 2 & Supplementary 3. Providing the data on a different scale between the main text and the figures is confusing; could you please provide also the data on the Figures on a normal scale?

Comment 5: Forest Plots (Figure 2, Figure 4 & Supplementary 3); could you please provide the "group at favor" on the "x_axis", on each side between the "vertical line of no-effect". For example, Forest Plot Figure_2: providing the side that favors "metastasis on the liver" vs the side that favors "no metastasis on the liver" on the "x_axis", could make the Figure easier to understand.

Comment 6: line 212, you report a "HR", but for the Figure (Supplementary data 3), you mention "relative risk". Could you please explain?

Comment 7: Despite the fact that data were from RCTs, the level of evidence seems still to be low. This meta-analysis reports outcomes with respect to the site(s) of metastasis, the treatment arms are not considered to be the comparisons in this particular case.The randomization of the original studies was performed with regard to the treatment. We are not considering two treatments comparisons in this meta-analysis, so randomization does not seem to provide security for the robustness of the outcome when considering metastatic site(s) as comparison. Could you please elaborate?

Comment 8: Patients enrolled in RCTs are strongly selected "PS 0-1". So the conclusions pertain more properly to patients in good PS and receiving the investigated chemo regimens; making it hard to make proper conclusions for the "real life" overall metastatic PC population. Could you please elaborate in the discussion? (and probably provide solutions, for example would a separate section analysis from large survival series of data from non randomized trials be of value?)

Reviewer #2: The authors evaluated the impact of metastatic site of pancreatic adenocarcinoma based on meta-analysis of prospective randomized studies. The liver was well- known to be most common metastatic site of pancreatic adenocarcinoma, and patients with disease progression usually died of hepatic failure. Other metastasis sites including the lung, lymph nodes or peritoneum should be included into analysis. Different treatment regimens were also found in the trials which may cause bias in evaluating the impact of metastatic sites on survival. The patient’s general performance and comorbidity will also influence the prognosis that cannot be assessed in this study. The results of this study did not provide additional information to our current knowledge to this disease.

6. PLOS authors have the option to publish the peer review history of their article (what does this mean?). If published, this will include your full peer review and any attached files.

Reviewer #1: No

Reviewer #2: No

---

## [Author Response · Author response to Decision Letter 0]

29 Jan 2020

Thank you very much for the review and suggestions.

Response to reviewers:

Reviewer#1: 

Comment 2: Is the HR of "liver metastasis" versus the "overall metastasis" (any kind considered) and/or "locally advanced" as comparator? This is not well stated in the manuscript. Could you please clearly state the comparator to the "liver metastasis". Also, considering that "locally advanced" patients are included, it probably would be fair to mention this in the limitations.

Response: Included the comparator in the methods (overall metastasis). Some trials included locally advanced disease, but we used as comparator just metastatic patients, we have included this statement in the limitations.

Comment 3: line 153; What do you mean by "PE" (is it Standard Error)? Also, on line 153 you mention that you estimate the "Confidence Interval limits", but you mention on line 148-149 that confidence intervals are supplied by the original articles. Why do say, that you you re-estimate the Confidence Intervals, could you please explain?

Response: We are sorry about this typo error, corrected in the line 153 (Standard error=SE). For the articles that the CI was available we used the value of the article, in the articles that CI was not available we calculated using the available data.

Comment 4: Why do you use different scales to report the "HRs & 95%CI", between the "main text" and the Forest Plots (Figure 2, Figure 4 & Supplementary 3). For the Forest Plots (Figure 2, Figure 4 & Supplementary 3) you use a "logarithmic scale" which is not stated for for Figure 2 & Supplementary 3. Providing the data on a different scale between the main text and the figures is confusing; could you please provide also the data on the Figures on a normal scale?

Comment 5: Forest Plots (Figure 2, Figure 4 & Supplementary 3); could you please provide the "group at favor" on the "x_axis", on each side between the "vertical line of no-effect". For example, Forest Plot Figure_2: providing the side that favors "metastasis on the liver" vs the side that favors "no metastasis on the liver" on the "x_axis", could make the Figure easier to understand.

Comment 6: line 212, you report a "HR", but for the Figure (Supplementary data 3), you mention "relative risk". Could you please explain?

Response: Sorry about the confuse forest plots, the graphs were remade on a non-logarithmic scale. We also included the information suggested to make it clearer.

The altered graphics:

- Comparison of metastasis in liver for overall survival - studies with HR

- Comparison of metastasis in liver for progression-free survival - studies with HR

- Comparison of studies with three sites of metastasis in relation to one site - studies with number of events.

The first two have a measure of HR effect because we were able to extract this information from the studies and use it in the meta-analysis. In the case of comparing the number of metastatic sites, the studies presented only the number of cases and not the hazard ratio, so we did the meta-analysis considering the proportion of events, the summary measure is relative risk.

Comment 7: Despite the fact that data were from RCTs, the level of evidence seems still to be low. This meta-analysis reports outcomes with respect to the site(s) of metastasis, the treatment arms are not considered to be the comparisons in this particular case. The randomization of the original studies was performed with regard to the treatment. We are not considering two treatments comparisons in this meta-analysis, so randomization does not seem to provide security for the robustness of the outcome when considering metastatic site(s) as comparison. Could you please elaborate?

Response: This was certainly one of the biggest limitations in the development of this project since there are no randomized data for metastasis in advanced pancreatic adenocarcinoma, our goal was to generate some solid evidence for the development of future studies based on this aspect. Regarding the treatment-related outcome, there is no impression that the treatments would objectively influence the results found, since no type of systematic treatment for advanced pancreatic cancer resulted in more than 10-12 months of median survival in any randomized trial. We included this information of randomization in the limitations of the study.

Comment 8: Patients enrolled in RCTs are strongly selected "PS 0-1". So the conclusions pertain more properly to patients in good PS and receiving the investigated chemo regimens; making it hard to make proper conclusions for the "real life" overall metastatic PC population. Could you please elaborate in the discussion? (and probably provide solutions, for example would a separate section analysis from large survival series of data from non-randomized trials be of value?)

Response: Included in the limitations of the study the PS in the RCT included and the need to analyze patient level data and large survival series data to draw better conclusions.

Reviewer #2: 

The authors evaluated the impact of metastatic site of pancreatic adenocarcinoma based on meta-analysis of prospective randomized studies. The liver was well- known to be most common metastatic site of pancreatic adenocarcinoma, and patients with disease progression usually died of hepatic failure. Other metastasis sites including the lung, lymph nodes or peritoneum should be included into analysis. Different treatment regimens were also found in the trials which may cause bias in evaluating the impact of metastatic sites on survival. The patient’s general performance and comorbidity will also influence the prognosis that cannot be assessed in this study. The results of this study did not provide additional information to our current knowledge to this disease.

Response: We cannot agree with the reviewer. The COMM-PACT consensus suggested stratification of metastatic pancreatic cancer in specific metastatic sites for all randomized studies to be developed in the future, our meta-analysis provides an objective analysis of the impact of metastatic site on disease outcomes, prospective studies that specifically assess this aspect do not exist. 

ACADEMIC EDITOR: 

Thank you very much for the review.

1. Please consider that the outcome from first-line and second-line studies might be different.

Response: We included just first-line trials in this meta-analysis. Line 113: Inclusion criteria were: (1) randomized prospective cohorts investigating first-line systemic treatment;

2. In figure 2, all included studies show no impact of survival, but the final HR was a positive impact of liver metastasis to the overall survival? How come? Please explain. Also, why the authors have Fuchs et al. 2015 and Fuchs et al. 2015.1. There is no explanation in the text and analysis. Do they refer to study arms 1 and 2? Please clearly describe them in the manuscript. 

Response: The articles included have negative results between comparison among treatments in the randomized trial, but the site of metastasis (liver versus other) confers worse outcomes among them. We included this limitation about the treatment arm in the discussion, after the reviewer 1 response. We are sorry about the explanation, Fuchs et al. 2015 and Fuchs et al. 2015.1. are each arm of the trial, we included an explanation in the results: 3.1 Liver Metastasis.

3. The legend of Figure 2 illustrates an HR 1.53 (1.15-2.02, p=0.003), which seems different from the figure itself (0.42[0.14-0.70]). Please explain. 

Response: Sorry about the confuse forest plots, the graphs were remade on a non-logarithmic scale. We also included the information suggested by the reviewer 1 to make it clearer.

4. There were no clear abbreviations in the legend for some short words in the figures. Please correct them. 

Response: Corrected as directed.

5. Liver metastasis includes huge liver (higher %) metastasis and tiny liver metastasis (lower % of occupying lesions). The study failed to exclude the factor, resulting in a not scientific enough conclusion. Please at least declare the limitations of the study. 

Response: Unfortunately, no randomized trial in metastatic pancreatic cancer considers volume of liver disease but only presence or absence, included this limitation in the discussion.

Response: Corrected as directed.

2. We noticed you have some minor occurrence(s) of overlapping text with the following previous publication(s), which needs to be addressed:

https://doi.org/10.1038/s41598-019-52334-y

https://doi.org/10.1016/j.clcc.2018.03.007

In your revision ensure you cite all your sources (including your own works), and quote or rephrase any duplicated text outside the Methods section. Further consideration is dependent on these concerns being addressed.

Response: Thanks for the guidance, corrected as directed (introduction). 

Response: Corrected as directed.

---

## [Decision Letter · Decision Letter 1]

21 Feb 2020

The impact of metastatic sites in advanced pancreatic adenocarcinoma, systematic review and meta-analysis of prospective randomized studies

PONE-D-19-32606R1

Dear Dr. Uson Junior,

We are pleased to inform you that your manuscript has been judged scientifically suitable for publication and will be formally accepted for publication once it complies with all outstanding technical requirements.

With kind regards,

Jason Chia-Hsun Hsieh, M.D. Ph.D

Academic Editor

PLOS ONE

Additional Editor Comments (optional):

All the questions were answered adequately.

Reviewers' comments:

Reviewer's Responses to Questions

**Comments to the Author**

1. If the authors have adequately addressed your comments raised in a previous round of review and you feel that this manuscript is now acceptable for publication, you may indicate that here to bypass the “Comments to the Author” section, enter your conflict of interest statement in the “Confidential to Editor” section, and submit your "Accept" recommendation.

Reviewer #1: All comments have been addressed

2. Is the manuscript technically sound, and do the data support the conclusions?

Reviewer #1: Yes

3. Has the statistical analysis been performed appropriately and rigorously? 

Reviewer #1: Yes

4. Have the authors made all data underlying the findings in their manuscript fully available?

Reviewer #1: Yes

5. Is the manuscript presented in an intelligible fashion and written in standard English?

Reviewer #1: Yes

6. Review Comments to the Author

Reviewer #1: (No Response)

7. PLOS authors have the option to publish the peer review history of their article (what does this mean?). If published, this will include your full peer review and any attached files.

Reviewer #1: No

---

## [Editor Report · Acceptance letter]

26 Feb 2020

PONE-D-19-32606R1 

The impact of metastatic sites in advanced pancreatic adenocarcinoma, systematic review and meta-analysis of prospective randomized studies 

Dear Dr. Uson Junior:

I am pleased to inform you that your manuscript has been deemed suitable for publication in PLOS ONE. Congratulations! Your manuscript is now with our production department. 

With kind regards,

on behalf of

Dr. Jason Chia-Hsun Hsieh 

Academic Editor

PLOS ONE